# Humidification Practices of Extremely Preterm Neonates: A Clinical Survey

**DOI:** 10.3390/healthcare10081437

**Published:** 2022-07-31

**Authors:** Nina Rizk, Carl D’Angio, Alison L. Kent

**Affiliations:** 1Department of Medicine, University of Rochester Medical Center, Rochester, NY 14642, USA; 2Division of Neonatology, Golisano Children’s Hospital, University of Rochester Medical Center, Rochester, NY 14642, USA; carl_dangio@urmc.rochester.edu; 3Department of Pediatrics, Golisano Children’s Hospital, University of Rochester Medical Center, Rochester, NY 14642, USA; alison_kent@urmc.rochester.edu; 4College of Health and Medicine, Australian National University, Canberra, ACT 2601, Australia

**Keywords:** neonatology, humidification, quality improvement

## Abstract

Extremely preterm neonates are at risk of morbidity and mortality related to their underdeveloped skin barrier. Humidified incubators are typically used in their care, but there is a paucity of literature to inform the standardization of specific evidence-based humidification practices in the NICU. A brief, voluntary, anonymous survey was distributed to our home institution and numerous national and international external institutions. Survey questions pertained to institutional humidification guidelines and were qualitatively analyzed. We received 89 responses from the home institution and 42 responses from the external institutions. Within the home institution, despite the presence of a guideline, individual practitioners reported varying practices in the starting levels of humidity and length of time spent in humidity. The results also demonstrated significant variability in individual humidification practices between the external institutions. There is no standard humidification guideline for extremely preterm neonates being cared for in the NICU. Further research is required to provide appropriate evidence on which to base clinical guidelines for the management of extremely preterm neonates to prevent morbidity and mortality in this population.

## 1. Introduction

During 2018, the rate of preterm birth in the United States rose for the fourth year in a row, with 1 in every 10 neonates born preterm [1,2]. Extremely preterm birth refers specifically to any birth prior to the completion of 28 weeks gestation. Approximately 50% of extremely preterm births result in mortality, and approximately 50% of survivors of extremely preterm birth experience significant morbidity [1,2]. Early morbidity associated with extreme preterm birth includes dehydration, hypothermia, electrolyte imbalances, and poor weight gain. These conditions are directly associated with immature physiologic development of the layers of the epidermis [3]. Extremely preterm neonates exhibit a high surface area to body mass ratio and an underdeveloped keratinized stratum corneum layer. Both of these characteristics are correlated with the subsequent risk of both heat loss and water loss through the highly permeable skin barrier. Absence of a robust protective skin barrier also places extremely preterm neonates at risk of infection and toxicity [4,5,6,7,8].

There is a strong consensus among healthcare professionals that humidification in and of itself is beneficial in the care of extremely preterm neonates [9,10,11,12]. The use of humidified incubators has been found to be the best method of preventing trans-epidermal heat/water loss in extremely preterm neonates by minimizing the water gradient between the internal environment of the neonate and the external environment of the incubator and, therefore, minimizing evaporation through the skin [9,10,11,12]. In addition, the use of humidified incubators minimizes the need for excessive fluid administration to prevent hypernatremia, hyperkalemia, and azotemia [13].

However, there is a lack of consensus among practitioners as to the best methods for using humidification in the care of these neonates. Studies have compared outcomes in neonates cared for with or without humidification, but no study has compared different percentages, durations, and weaning practices of humidification to one another in a controlled manner [9,10,11,12]. As a result, different institutions employ different practices and there is no clear standard of care for humidification methods. Without this evidence, it is unclear how best to care for this vulnerable population of newborns in order to minimize mortality and morbidity.

The aims of this international survey were to determine current humidification practices in the care of extremely preterm neonates, the presence of institutional guidelines, and evaluate within one unit adherence to an institutional guideline.

## 2. Materials and Methods

This was a cross sectional study utilizing an electronic survey via SurveyMonkey^®^ (San Mateo, CA, USA). The survey was distributed to healthcare providers in the USA and international institutions. This voluntary and anonymous survey targeted healthcare providers at varying levels of practice, including physicians (MDs, DOs), advanced practice providers (APPs), nurse practitioners (NPs), and nurses (RNs), and any other healthcare providers who work in the setting of a NICU. In order to recruit a diverse range of survey respondents, the survey was sent via email to healthcare providers at Golisano Children’s Hospital at the University of Rochester Medical Center (GCH/URMC), institutions involved with the Neonatal Kidney Collaborative (NKC), and practitioner members of the National Association of Neonatal Nurses (NANN). The study was approved by the University of Rochester’s Institutional Review Board (Study 00004448).

GCH/URMC (the home institution of the study authors in Rochester, NY) is a Level IV NICU with 68 beds, staffed by 17 neonatologist physicians, 40 advanced practice providers, and 250 nurses. Distribution of the survey was approved by the clinical leadership of the NICU. Following approval, the survey link was emailed to all staff members, and was available for a total of three weeks with one reminder sent after one week. The survey responses from GCH/URMC provided information regarding the variability in humidification practices within a single institution, in the presence of a unit guideline.

The NKC includes more than 100 members (neonatologists and nephrologists) in USA and international institutions and was established to improve newborn kidney health through collaborative research, advocacy, and education. Distribution of the survey was approved by the NKC Steering Committee. The survey link was emailed to neonatal members and was available for a total of three weeks with a reminder at one week. The NANN has nearly 8000 members practicing in US institutions and was established to shape neonatal nursing through collaborative research, practice, education, and professional development. Distribution of the survey was approved by the NANN Research Committee. Following approval, the survey link was posted in an online forum, and was available for a total of three weeks. The survey responses from the NKC and the NANN provided information regarding the variability in humidification methods across institutions.

The survey began with a statement of consent, a brief introduction, and included a total of fifteen questions (Appendix A). The first three questions asked for indirect identifiers: country of origin, institution of origin, and professional status. The following twelve questions related to humidification practices in the NICU, including specific details regarding percentage of humidification, duration of humidification, weaning of humidification, administration of fluids, and candida prophylaxis.

The survey responses were then analyzed using descriptive statistics and qualitative analysis. This research received no specific grant from any funding agency in the public, commercial, or not-for-profit sectors.

## 3. Results

A total of 89 responses were received from GCH/URMC and 42 responses from external institutions (15 NKC and 27 NANN). Survey respondents represented at least four distinct professional categories (Table 1) and more than thirty institutions located in five countries, sixteen states, and Washington DC (Table 2). All survey respondents indicated that their institution has a guideline in place for the use of humidification in the NICU.

Responses from GCH/URMC demonstrated significant variability, despite the existence of a standard guideline. The GCH/URMC guideline defined neonates requiring humidification as those who weigh less than one thousand grams at birth. The guideline suggests a starting percent humidification of 70%; however, less than 5% of respondents selected this option in the survey. The majority of respondents (44%) selected a starting percent humidification of at least 90%. The guideline recommends an ending percent humidification of at least 50% and duration of humidification for 1 week, with 63% of and 83% of the respondents selecting these options, respectively.

The current guideline at GCH/URMC is less explicit in its recommendations for humidification weaning, stating only that ELBW neonates should remain in a humidified incubator for up to one week before being transferred to a non-humidified incubator. The guideline does not include information about the initiation of weaning or the increments of weaning. When asked for the postnatal age at which humidification weaning should begin, 38% selected 3 days. Fifteen percent stated that the decision to wean humidification was dependent on various factors, including gestational age, fluid status, and electrolyte balance. When asked for the percent intervals at which humidification weaning should occur, 89% selected 10%, demonstrating a strong consensus despite the absence of a specific directive.

The questions pertaining to the administration of intravenous fluids, which allowed for stratification of neonates by gestational age, demonstrated similar variability. The guideline at GCH/URMC does not explicitly list a recommended rate for the administration of fluids to the neonate, and this ambiguity is reflected in the responses. For neonates with a gestational age of less than 25 weeks, 35% answered that they administer fluids at 100 mL/kg/day and 35% at 90 mL/kg/day. For neonates with a gestational age between 26 weeks and 28 weeks, 42% answered that they administer fluids at 90 mL/kg/day and 37% at 80 mL/kg/day. Finally, although there is no mention of microbial prophylaxis in the guideline, there was agreement among respondents that candida prophylaxis is not routinely administered for extremely preterm neonates.

These differences in humidification practices were also noted in the survey responses from external institutions nationally and internationally. When asked about incubator maintenance, 50% of respondents indicated that they changed the incubators with humidification weekly. However, 24% indicated they followed some other practice, either changing the incubator only when humidification was completely discontinued (7%), changing the incubator with varying frequency dependent on the percentage of humidification being used within the incubator (7%), or changing the incubator in accordance with some other institutional or manufacturer guideline (10%).

In total, 48% of the respondents indicated that they began humidification for neonates born at or before 28 weeks gestation. Forty-three percent followed some other practice, either starting humidification for neonates born at some later gestational age (21%), starting humidification for neonates born below a specific birth weight (10%), or starting humidification with a combined consideration of both gestational age and birth weight (12%). In total, 45% indicated that they began humidification between 80% and 89%. Fourteen percent indicated that they followed some other practice, citing gestational age as the most important factor in determining initial humidification percentage.

Fifty-seven percent of respondents indicated that they weaned humidification starting at a postnatal age of seven days. Twenty-two percent of respondents indicated that they followed some other practice, either weaning humidification at some later postnatal age (5%), or weaning humidification based on some other criteria such as weight, temperature, skin integrity, or overall health (17%). Related to the percentage increments of weaning, 41% of respondents indicated that they weaned by increments of 5%. Thirty-three percent indicated they followed some other practice, weaning humidification by some other percentage increment.

In total, 14% indicated that they discontinued humidification at a postnatal age of 7 days and 14% at a postnatal age of 10 days. Seventy-one percent reported that they followed some other practice, either ending humidification at a particular percent humidification (5%), ending humidification at a particular later postnatal age (45%), or ending humidification based on other criteria such as gestational age, current weight, temperature, or overall health (21%). In total, 24% of the respondents discontinued humidification in the incubator at 50% or higher, 26% between 40% and 49%, and 31% at 39% or lower. Survey responses related to the primary outcomes measured are provided in Table 3.

Finally, for neonates with a gestational age of less than 25 weeks, 48% of the respondents answered that they commence fluids at 100 mL/kg/day and 26% at 80 mL/kg/day. Twenty-one percent of the respondents either listed an alternative rate or range of rates (10%), other criteria that determined the rate such as provider preference (2%) or clinical presentation (2%), or simply that variation existed (7%). For neonates with a gestational age between 26 weeks and 28 weeks, 41% of the respondents commenced fluids at 80 mL/kg/day and 38% at 100 mL/kg/day. Fourteen percent provided an alternative rate or range of rates, other criteria that determined the rate such as provider preference (2%) or clinical presentation (2%), or simply that variation existed (2%).

Unlike the results from the home institution, those from the external institutions showed greater variation with respect to candida prophylaxis. In total, 74% did not routinely administer candida prophylaxis treatment during humidification while 12% did. The remaining 14% responded that this decision was reliant upon either provider preference (2%), positive rectal yeast swab (2%), neonatal birthweight (5%), or presence of an intravenous line (5%). Survey responses related to the secondary outcomes measured are provided in Table 4.

## 4. Discussion

This study has shown marked variation in clinical practice, both within and between institutions and countries, for incubator humidification in extremely preterm neonates. Until a recent systematic review, there has been a lack of research regarding humidification practices for preterm infants, making it difficult to develop an evidence-based guideline and a standardized practice [14]. This may result in increased morbidity and mortality among these vulnerable patients, primarily related to loss of water and heat through an immature and underdeveloped skin barrier.

The survey responses from our single institution (GCH/URMC) demonstrated inconsistent practice despite the presence of a unit guideline. The survey responses suggest that the guideline is either poorly understood, not utilized, or has not been updated to reflect consensus in clinical practice given the paucity of evidence. In the absence of a specific recommendation for humidification weaning, survey respondents provided an even greater variety of answers, claiming that much of what they do is based on clinical judgement related to individual patient factors. Our literature review did not find any national guidelines on humidification practice; however, institutional guidelines could be found. The paucity of evidence on the percentage of starting, weaning, and cessation of humidification obviously drives the variation in practice within units, between units, and between countries.

This study was an observational cross-sectional survey, and, as such, it has limitations. The data was collected via an online link and, therefore, only those with access to an electronic device and the internet were able to access the survey. The survey was distributed to healthcare practitioners at GCH/URMC, the home institution of the authors. Despite the survey being anonymous and voluntary, survey recipients may have been influenced by the fact that their colleagues would be collecting and reviewing the results. The survey was also distributed to a limited group of external healthcare practitioners (NKC and NANN). As a result, only those individuals who were members of these organizations had access to the survey, and so our assessment of national and international institutions was limited to those whose employees were members of the included organizations. We were limited by a small subject group, only receiving 42 responses from external institutions. Finally, because the survey was distributed via listservs, we were unable to determine whether our survey results are representative of the sample because we could not calculate the percentage of respondents in relation to the total number of participants surveyed. However, given the wide range of results from these external institutions, it is likely that we would have continued to have found wide variations in practice. 

In their systematic review, Glass and Valdez [14] reached a similar conclusion to ours in that there are differing opinions on incubator humidity levels, and that recommendations for the humidification of preterm infants have varied in parallel with the development of new and improved humidity technologies. Further, they pointed out that there has been inconsistent use of humidification in the clinical setting, a paucity of research addressing specific patient outcomes, and a lack of large clinical trials to support any one conclusion. The study in question ultimately found that high levels of humidity in the first five days of life for preterm infants might impede skin maturation and subsequently promote trans-epidermal water loss. However, even after their review, the authors noted that “future studies are needed comparing incubator humidity levels and duration correlated with gestational age” [14].

The reason that humidification of neonates is so widely employed is because it reduces morbidity, including dehydration, hypothermia, electrolyte imbalances, and poor weight gain, associated with extremely preterm birth. Given the disparity in practice within and between institutions, further research is required to determine the best humidification practices to minimize the morbidity associated with extreme prematurity. An institutional review of current practice is now underway, which will evaluate humidification percentages at commencement, weaning, and cessation, assessing the influence on temperature control, electrolyte imbalance, and weight loss. Following this a new guideline will be formulated and following an education program delivered to all staff, data will be collected again to determine whether the new guideline results in a consistent approach in clinical care and improved temperature, electrolyte balance, and reduced weight loss.

## Figures and Tables

**Table 1 healthcare-10-01437-t001:** Professional distribution of respondents.

	External Institutions (NKC) N = 15	External Institutions (NANN) N = 27	Home Institution (GCH/URMC) N = 89
MD	11	0	8
NP	1	1	11
RN	2	15	67
APP	0	8	3
Other	1	3	0

MD (Medical Doctor), NP (Nurse Practitioner), RN (Registered Nurse), APP (Advanced Practice Provider), NKC (Neonatal Kidney Collaborative), NANN (National Association of Neonatal Nurses), and GCH/URMC (Golisano Children’s Hospital/University of Rochester Medical Center).

**Table 2 healthcare-10-01437-t002:** Geographic distribution of respondents.

Country of Origin	Number of Institutions
Australia	1
Canada	1
Czech Republic	1
India	1
United States	35
**State of Origin**	**Number of Institutions**
Arkansas	1
Arizona	1
Colorado	3
Connecticut	1
Georgia	2
Illinois	1
Massachusetts	1
Missouri	1
New Jersey	1
New York	5
North Carolina	1
Ohio	4
Oklahoma	1
Pennsylvania	2
Texas	2
Virginia	3
Washington DC	1
Unknown	4

**Table 3 healthcare-10-01437-t003:** Survey responses related to primary outcomes.

Answers	Outside Institutions (NKC = 15/NANN = 27) N = 42	Home Institution (GCH/URMC) N = 89
How often do you change the incubators with humidification?		
Every 1 Week	21 (50%)	72 (81%)
Every 2 Weeks	11 (26%)	2 (2%)
Other	10 (24%)	15 (17%)
No Response	0 (0%)	0 (0%)
At what gestational age do you begin humidification?		
</= 28 Weeks	20 (48%)	52 (58%)
</= 27 Weeks	2 (5%)	10 (11%)
</= 26 Weeks	1 (2%)	5 (6%)
</= 25 Weeks	1 (2%)	5 (6%)
Other	18 (43%)	16 (18%)
No Response	0 (0%)	1 (1%)
At what percentage do you begin humidification?		
90% or Higher	2 (5%)	39 (44%)
80% to 89%	19 (45%)	30 (34%)
70% to 79%	13 (31%)	4 (4%)
60% to 69%	2 (5%)	1 (1%)
50% to 59%	0 (0%)	2 (2%)
49% or Lower	0 (0%)	0 (0%)
Other	6 (14%)	13 (15%)
No Response	0 (0%)	0 (0%)
At what postnatal age do you wean humidification?		
1 Day	1 (2%)	8 (9%)
3 Days	2 (5%)	34 (38%)
5 Days	5 (12%)	19 (21%)
7 Days	24 (57%)	14 (16%)
Not Applicable	1 (2%)	1 (1%)
Other	9 (22%)	13 (15%)
No Response	0 (0%)	0 (0%)
At what percentage increments do you wean humidification?		
5%	17 (41%)	4 (4%)
10%	9 (21%)	79 (89%)
Not Applicable	2 (5%)	0 (0%)
Other	14 (33%)	5 (6%)
No Responses	0 (0%)	1 (1%)

**Table 4 healthcare-10-01437-t004:** Survey responses related to secondary outcomes.

Answers	Outside Institutions (NKC = 15/NANN = 27) N = 42	Home Institution (GCH/URMC) N = 89
Do you routinely administer candida prophylaxis treatment during humidification?		
Yes	5 (12%)	0 (0%)
No	31 (74%)	87 (98%)
Other	6 (14%)	2 (2%)
No Response	0 (0%)	0 (0%)
At what rate do you administer intravenous fluids during the first 24 h for neonates ≤ 25 weeks gestation?		
70 mL/kg/day	0 (0%)	11 (12%)
80 mL/kg/day	11 (26%)	11 (12%)
90 mL/kg/day	2 (5%)	31 (35%)
100 mL/kg/day	20 (48%)	31 (35%)
Other	9 (21%)	3 (3%)
No Response	0 (0%)	2 (2%)
At what rate do you administer intravenous fluids during the first 24 h for neonates 26-28 weeks gestation?		
70 mL/kg/day	0 (0%)	9 (10%)
80 mL/kg/day	17 (41%)	33 (37%)
90 mL/kg/day	3 (7%)	37 (42%)
100 mL/kg/day	16 (38%)	7 (8%)
Other	6 (14%)	1 (1%)
No Response	0 (0%)	2 (2%)

## Data Availability

The data presented in this study are available in Table 1, Table 2, Table 3 and Table 4.

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
