# Peer review of "Humidification Practices of Extremely Preterm Neonates: A Clinical Survey"

_healthcare, 2022, doi:10.3390/healthcare10081437_

Round 1
Reviewer 1 Report
This survey to healthcare providers aimed to determine the current humidification practices in the care of extremely preterm neonates.
This is an interesting objective but the manuscript has some flaws that should be addressed.
Queries:
The statement (line 190-191) “The lack of research regarding humidification practices…” is inaccurate, as the authors ignored an important recent systematic review on preterm infant incubator humidity levels (Glass 2021). Therefore, this statement should be reformulated and the results of this systematic review should be appropriately addressed and compared with results from this survey.
To get an idea of whether the survey results are representative of the sample, the percentage of respondents should be specified in relation to the total number of participants surveyed.
The term 'isolette' is used loosely in the manuscript as a synonym for closed incubator. In fact, Isolette is a Drager brand and in some surveyed units, different brands of incubators may have been used. Therefore, throughout the manuscript, 'isolette' should be replaced by 'incubator'.
Abbreviations in Table 1 should be explained in full in the table footnote.
In the questionnaire, only Candida prophylaxis was enquired regarding infection. Therefore, were it is stated ‘prevention of infection’ (line 96) it should be replaced by ‘Candida prophylaxis’.
Reference
Glass L, Valdez A. Preterm infant incubator humidity levels: a systematic review. Adv Neonatal Care. 2021 Aug 1;21(4):297-307. doi: 10.1097/ANC.0000000000000791.
Reviewer 2 Report
Congratulations to the authors of this beautiful interview/cross sectional study. I read with great attention and pleasure from the first to the last word. For the first time, since I’ve been reviewer, I have almost nothing to suggest. I just think there’s some minor punctuation error, so, correct it.
Congratulations again!
Round 2
Reviewer 1 Report
The questions were clarified and the revised manuscript is improved.